

# Disinfection efficacy of sodium hypochlorite and glutaraldehyde and their effects on the dimensional stability and surface properties of dental impressions: a systematic review

Yuan Qiu*, Jiawei Xu*, Yuedan Xu, Zhiwei Shi, Yinlin Wang, Ling Zhang and Baiping Fu

Stomatology Hospital, School of Stomatology, Zhejiang University School of Medicine, Dental Materials and Devices for Zhejiang Provincial Engineering Research Center, Zhejiang Provincial Clinical Research Center for Oral Diseases, Key Laboratory of Oral Biomedical Research of Zhejiang Province, Cancer Center of Zhejiang University, Hangzhou, China
* These authors contributed equally to this work.

## ABSTRACT

**Objective**. To systematically evaluate the disinfection efficacy of the two most frequently used disinfectants, sodium hypochlorite and glutaraldehyde, and their effects on the surface properties of four different dental impression materials.

**Methods**. A systematic literature search was performed in four databases until May 1st, 2022 to select the studies which evaluated disinfection efficacy of disinfectants or surface properties of dental impressions after chemical disinfection.

**Main results**. A total of 50 studies were included through electronic database searches. Of these studies, 13 studies evaluated disinfection efficacy of two disinfectants, and 39 studies evaluated their effects on the surface properties of dental impressions. A 10-minute disinfection with 0.5–1% sodium hypochlorite or 2% glutaraldehyde was effective to inactivate oral flora and common oral pathogenic bacteria. With regard to surface properties, chemical disinfection within 30 min could not alter the dimensional stability, detail reproduction and wettability of alginate and polyether impressions. However, the wettability of addition silicone impressions and the dimensional stability of condensation silicone impressions were adversely affected after chemical disinfection, while other surface properties of these two dental impressions were out of significant influence.

**Conclusions**. Alginate impressions are strongly recommended to be disinfected with 0.5% sodium hypochlorite using spray disinfection method for 10 min. Meanwhile, elastomeric impressions are strongly recommended to be disinfected with 0.5% sodium hypochlorite or 2% glutaraldehyde using immersion disinfection method for 10 min, however, polyether impression should be disinfected with 2% glutaraldehyde.

Corresponding authors
Ling Zhang, jorlinzhang@zju.edu.cn
Baiping Fu, fbp@zju.edu.cn

## INTRODUCTION

The oral cavity is the entry portal of the gastrointestinal tract and comprised of many surfaces (*Aas et al., 2005*; *Sampaio-Maia et al., 2016*). Each of them is coated with a plethora of microorganisms, which made up the proverbial biofilm (*Aas et al., 2005*; *Yamashita & Takeshita, 2017*). Oral biofilms harbor more than 700 species of bacteria (*e.g.*, *Staphylococcus aureus*, *Pseudomonas aeruginosa*, *Bacillus atrophaeus*) as well as fungi and viruses, which might cause dental and periodontal infections (*Chen et al., 2010*; *Doddamani, Patil & Gangadhar, 2011*; *Maddi & Scannapieco, 2013*). Previous studies have reported that oral infections were implicated in the etiopathogenesis of several important chronic systemic diseases (*Chen et al., 2010*; *Maddi & Scannapieco, 2013*).

Dental impression materials are widely used in dentistry for making accurate casts of the dentition and its neighboring oral tissues, capable of recording the prepared tooth and its surrounding anatomic topography of the desired area (*Davis & Powers, 1994*; *Khinnavar, Kumar & Nandeeshwar, 2015*; *Martins et al., 2017*; *Nimonkar et al., 2019*; *Piva et al., 2019*; *Saleh Saber, Abolfazli & Kohsoltani, 2010*). During the dental impression procedure, the dental impression materials come into contact with blood and saliva, which contain potentially pathogenic micro-organisms and some viruses (*Azevedo et al., 2019*). Thus, in the absence of immediate disinfection, dental impressions contaminated with saliva or blood can serve as a source of cross infection between patients and dental care providers (*AlZain, 2019*; *Amin et al., 2009*; *Carvalhal et al., 2011*; *Drennon, Johnson & Powell, 1989*; *Guiraldo et al., 2018*; *Karaman, Oztekin & Tekin, 2020*; *Khatri et al., 2020*; *Lad et al., 2015*; *Shetty, Kamat & Shetty, 2013*).

The dental impressions should possess the properties of keeping stable dimensions and reproducing tiny details of the dentition and its surrounding oral tissues, which are termed as dimensional stability and detail reproduction separately (*Guiraldo et al., 2017*; *Guiraldo et al., 2012*). Meanwhile, it is of vital importance for dental impressions to keep their surface moist and smooth (*Karaman, Oztekin & Tekin, 2020*; *Lad et al., 2015*). The surface properties of the dental impressions might be altered during impression disinfection procedures, deteriorating the surface properties of the corresponding casts (*Amin et al., 2009*; *Carvalhal et al., 2011*). Thus, a desired disinfectant must be an effective antimicrobial agent, and in the meanwhile has no adverse effect on the impression accuracy, impression surface properties and the corresponding casts (*Taylor, Wright & Maryan, 2002*).

The immediate disinfection of the dental impressions has been considered as a routine procedure in dental clinics and dental laboratories (*Hamedi Rad, Ghaffari & Safavi, 2010*; *Hiraguchi et al., 2012*; *Karaman, Oztekin & Tekin, 2020*; *Lepe et al., 2002*; *Shetty, Kamat & Shetty, 2013*; *Silva & Salvador, 2004*; *Thouati et al., 1996*). As physical disinfections might result in temperature rise, which could cause measurable deformations in the molds (*Carvalhal et al., 2011*; *Silva & Salvador, 2004*). Chemical disinfectants might be more suitable for dental impression materials because they are not only widely used, but also easily managed in dental clinics. Multifarious chemicals have been used to disinfect the dental impressions, such as sodium hypochlorite, glutaraldehyde, chlorhexidine, iodophor, peracetic acid and mixed disinfectants (*Amin et al., 2009*; *Guiraldo et al., 2017*; *Herrera*

*& Merchant, 1986*). Of these, sodium hypochlorite and glutaraldehyde are probably the most frequently used disinfectants (*AlZain, 2019*; *Babiker, Khalifa & Alhajj, 2018*; *Khinnavar, Kumar & Nandeeshwar, 2015*; *Martin, Martin & Jedynakiewicz, 2007*; *Walker et al., 2007*). Since 1988, the American Dental Association (ADA) has recommended immersion with sodium hypochlorite of all impression materials for the manufacturer's recommended contact time (no more than 30 min) (*American Dental Association, 1988*; *American Dental Association, 1992*; *American Dental Association, 1996*). Recently, medical organizations in various countries and regions have proposed guidelines related to infection prevention and control (*Australian Dental Association, 2021*; *of Health, 2019*; *of Saudi Arabia Ministry of Health, 2018*; *Nidhi et al., 2021*). According to these guidelines, impressions are recommended to be disinfected with 0.5% sodium hypochlorite or 2% glutaraldehyde for 10 min using spray or immersion disinfection method.

Previous studies have evaluated the effect of various disinfectants on surface properties of dental impressions (*AlZain, 2019*; *Hiraguchi et al., 2013*; *Martins et al., 2017*; *Rentzia et al., 2011*). Unfortunately, there were controversial findings of the dental impression disinfection because different disinfectants and different disinfecting procedures were adopted (*AlZain, 2019*; *Hiraguchi et al., 2013*; *Lad et al., 2015*; *Martins et al., 2017*; *Rentzia et al., 2011*; *Shetty, Kamat & Shetty, 2013*). Meanwhile, it is necessary to re-evaluate their effects on surface properties of the dental impressions after following the available impression disinfection guidelines (*American Dental Association, 1988*; *American Dental Association, 1992*; *American Dental Association, 1996*; *Australian Dental Association, 2021*; *of Health, 2019*; *of Saudi Arabia Ministry of Health, 2018*; *Nidhi et al., 2021*). Therefore, the purpose of this systematic review was to evaluate the disinfection efficacy of frequently-used disinfectants, such as sodium hypochlorite and glutaraldehyde, as well as their effect on the dimensional stability, detail reproduction and surface properties of different impression materials. This will provide overall guidance for establishing standardized disinfection procedures of the dental impressions.

## MATERIALS AND METHODS

### Data source and search strategy

A systematic literature search was conducted on Cochrane, PubMed, EMBASE and Web of Science until May 1st, 2022. The detailed search strategy is presented in Table 1. The MeSH terms and key words were listed as follows: dental impression materials, disinfection, disinfectant, efficacy, surface properties, dimensional stability, detail reproduction, wettability. References of relevant articles and reviews were also searched for additional eligible studies potentially missed.

The study protocol was registered in the International Prospective Register of Systematic Reviews (PROSPERO: CRD42021285551). The Preferred Reporting Items for Systematic Reviews and Meta-Analyses (PRISMA) guidelines were applied whenever applicable (*Knobloch, Yoon & Vogt, 2011*).
**Table 1  Search strategies for four databases.**

(1) The search strategy developed for PubMed
((Dental Impression Materials[MeSH Terms]) OR (Alginate) OR (polyether*) OR (addition silicone*) OR (condensation silicone*))AND ((Disinfection) OR (Disinfectant) OR (Sodium Hypochlorite) OR (glutaraldehyde) OR (Immersion) OR (spray)) AND ((Surface Properties[MeSH Terms]) OR (dimensional stability[Text Word]) OR (stability) OR (detail reproduction[Text Word]) OR (Wettability) OR (efficacy) OR (Staphylococcus aureus[Text Word]) OR (Pseudomonas aeruginosa[Text Word]) OR (Candida Albicans[MeSH Terms]))

(2) The search strategy developed for Web of Science
(("Dental Impression" OR Dental Impressions" OR "Alginate" OR "Polyether" OR "Condensation Silicone" OR "Addition Silicone") AND ("Disinfection" OR "Disinfectant" OR "Disinfectants" OR "Sodium Hypochlorite" OR "Glutaraldehyde" OR "Immersion" OR "Spray") AND ("Surface Property" OR "Surface Properties" OR "dimensional stability" OR "detail reproduction" OR " wettability" OR "efficacy" OR "Staphylococcus aureus" OR "Pseudomonas aeruginosa" OR "Candida Albicans"))

(3) The search strategy developed for EMBASE
('dental materials'/exp OR 'elastomer'/exp OR 'alginic acid'/exp OR 'condensation silicone':ti,ab,kw OR 'additional silicone':ti,ab,kw OR polyether:ti,ab,kw) AND ('disinfection'/exp OR 'disinfectant agent'/exp OR 'hypochlorite sodium'/exp OR 'glutaraldehyde'/exp OR 'immersion'/exp OR spray:ti,ab,kw) AND ('surface property'/exp OR 'dimensional stability':ti,ab,kw OR 'detail reproduction':ti,ab,kw OR 'wettability'/exp OR 'efficacy':ti,ab,kw OR 'Staphylococcus aureus':ti,ab,kw OR 'Pseudomonas aeruginosa':ti,ab,kw OR 'Candida Albicans':ti,ab,kw) AND [embase]/lim

(4) The search strategy developed for the Cochrane Library
#1 MeSH descriptor: [Dental Impression Materials] explode all trees
#2 MeSH descriptor: [Silicone Elastomers] explode all trees
#3 MeSH descriptor: [Alginates] explode all trees
#4 Polyether*
#5 Addition Silicone*
#6 Condensation Silicone*
#7 #1 OR #2 OR #3 OR #4 OR #5 OR #6
#8 MeSH descriptor: [Disinfection] explode all trees
#9 MeSH descriptor: [Dental Disinfectants] explode all trees
#10 MeSH descriptor: [Sodium Hypochlorite] explode all trees
#11 MeSH descriptor: [Glutarates] explode all trees
#12 MeSH descriptor: [Immersion] explode all trees
#13 Spray
#14 #8 OR #9 OR #10 OR #11 OR #12 OR #13
#15 MeSH descriptor: [Surface Properties] explode all trees
#16 dimensional stability
#17 detail reproduction
#18 MeSH descriptor: [Wettability] explode all trees
#19 efficacy
#20 Staphylococcus aureus
#21 Pseudomonas aeruginosa
#22 Candida Albicans
#23 #16 OR #17 OR #18 OR #19 OR #20 OR #21 OR #22
#24 #7 AND #14 AND #23

## Study selection criteria

To be eligible, studies had to satisfy all the following inclusion criteria: (1) laboratory (*in vitro*) studies; (2) studies based on impression materials frequently used in dentistry, including alginate, polyether, condensation silicone or addition silicone; (3) studies investigated sodium hypochlorite and glutaraldehyde as disinfectants. The disinfection methods included spray and immersion; (4) studies performed accurate evaluation of efficacy of the two disinfectants, or investigated surface properties of the dental impressions before and after disinfection, including dimensional stability, detail reproduction and wettability; (5) studies were published in English and Chinese.

## Study exclusion criteria

The studies that satisfied one of the following exclusion criteria were excluded: (1) *in vivo* studies; (2) studies investigated on disinfectants other than sodium hypochlorite or glutaraldehyde; (3) studies based on ambiguous disinfection procedures; (4) studies based on physical disinfection methods, including ultraviolet or microwave disinfection; (5) studies did not report their measurement results; (6) studies did not have control groups; (7) studies were not reported in English and Chinese.

The systematic literature search was carried out independently by two authors (Qiu Y, Xu J). If there were any disagreements, the study would be re-evaluated by a third investigator (Xu Y).

## Data extraction

Different impression materials, concentrations of different disinfectants, disinfection methods, disinfection duration, surface properties and the respective results in the articles were extracted.

Data extraction was performed independently by two reviewers (Qiu Y, Xu J) and any discrepancies were resolved by consensus.

## Assessment of risk of bias

To evaluate the methodological rigor of the included studies, the Joanna Briggs Institute (JBI) Critical Appraisal Checklist tool was adopted, including nine evaluation items (*The Joanna Briggs Institute, 2016*). Each evaluation item was rated as "yes", "no", "unclear", or "not applicable". Each study was graded according to the "yes" scores. Scores of 1–3 "yes" were considered as the high risk of bias, scores of 4–6 "yes" were the moderate risk of bias and scores of 7–9 'yes' were the low risk of bias.

Included studies were assessed and scored independently by two reviewers (Qiu Y, Xu J) and any disagreements were resolved by consensus.

# RESULTS

## Study selection

In total, the electronic search identified 2,044 articles from four databases (PubMed/MED-LINE, Web of Science, Embase and Cochrane Library). After duplicates had been removed, 1,825 articles were obtained. After titles and abstracts had been analyzed, 107 articles
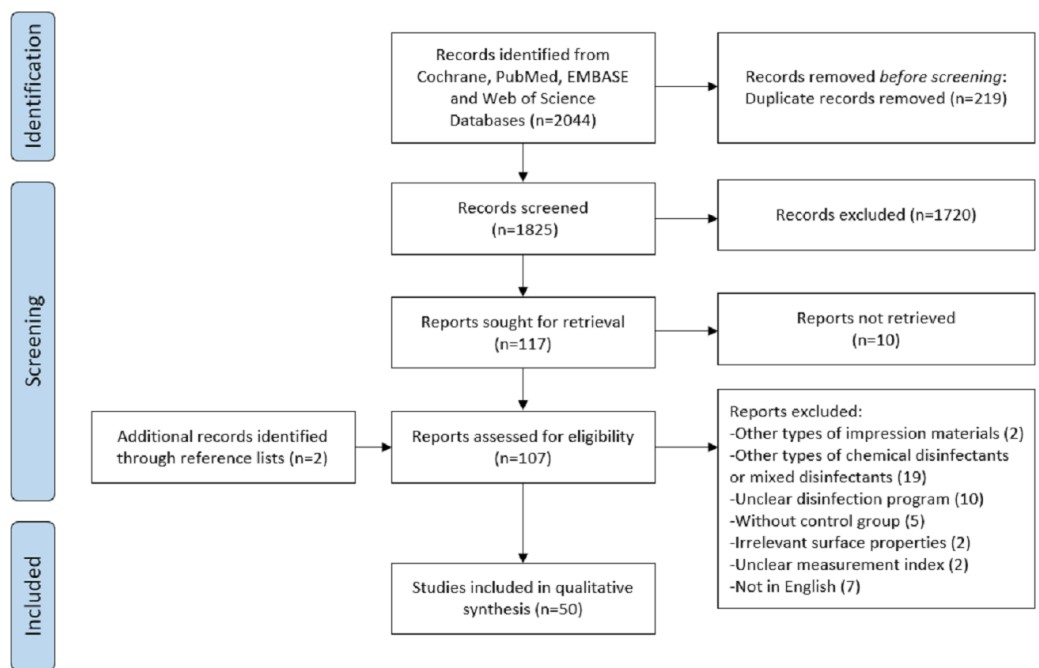

**Figure 1 Flowchart of the systematic review.**

remained. After full texts were scrutinized, 50 studies were included in the analysis. Detailed study selection was presented in Fig. 1.

Of the included studies, the most commonly used disinfection method was immersion followed by spray. The concentrations of the two disinfectants, sodium hypochlorite and glutaraldehyde, ranged from 0.05% to 5.25% and 0.13% to 2.5%, respectively. In addition, the disinfection duration ranged from 30 s to 24 h. To evaluate the disinfection efficacy of two disinfectants, the dental impressions were made directly from patients, volunteers (*Aeran et al., 2015*; *Azevedo et al., 2019*; *Haralur et al., 2012*; *Jeyapalan et al., 2018*; *Rajendran et al., 2021*; *Westerholm2nd et al., 1992*), or they were contaminated with specific bacteria after impression making (*Badrian et al., 2012*; *Choudhury et al., 2018*; *Doddamani, Patil & Gangadhar, 2011*; *Mendonca et al., 2013*; *Rentzia et al., 2011*; *Schwartz et al., 1994*; *Taylor, Wright & Maryan, 2002*; *Westerholm2nd et al., 1992*). When the impression surface properties were evaluated, stainless test blocks as described following ADA Specification or ISO Standards (*Abdelaziz, Hassan & Hodges, 2004*; *Amalan, Ginjupalli & Upadhya, 2013*; *Azevedo et al., 2019*; *Carvalhal et al., 2011*; *Davis & Powers, 1994*; *Gounder & Vikas, 2016*; *Guiraldo et al., 2017*; *Guiraldo et al., 2012*; *Khatri et al., 2020*; *Khinnavar, Kumar & Nandeeshwar, 2015*; *Langenwalter, Aquilino & Turner, 1990*; *Martins et al., 2017*; *Nassar & Chow, 2015*; *Nassar et al., 2017*; *Shambhu & Gujjari, 2010*; *Silva & Salvador, 2004*; *Thouati et al., 1996*; *Walker et al., 2007*; *Yilmaz et al., 2007*), as well as typical metal or acrylic maxillary or mandible dentition models were used for impression making (*Adabo et al., 1999*; *Babiker, Khalifa & Alhajj, 2018*; *Herrera & Merchant, 1986*; *Lepe & Johnson, 1997*; *Nimonkar et al., 2019*; *Rentzia et al., 2011*; *Rueggeberg et al., 1992*;

*Sinobad et al., 2014*). Several studies used specially designed model, including stainless steel model with a dental replica of two teeth prepared for complete crowns (*Drennon, Johnson & Powell, 1989*; *Saleh Saber, Abolfazli & Kohsoltani, 2010*), cylindrical stainless steel model (*Hiraguchi et al., 2013*; *Piva et al., 2019*; *Queiroz et al., 2013*), cuboid metallic model (*Hamedi Rad, Ghaffari & Safavi, 2010*) and ridge-shaped epoxy resin model (*Hiraguchi et al., 2012*). Due to lack of the standardized and identical experiment designs of the included studies, meta-analysis could not be carried out. The results of retrieved studies evaluating disinfection efficacy of disinfectants and surface properties of impressions were presented separately in Tables 2–7.

## Risk of bias assessment

The estimated risks of bias of included studies were summarized in Fig. 2. In general, 46 studies were at a low risk of bias, and four studies were at a moderate risk of bias. Two studies did not pour dental stone into alginate impressions of control group immediately after impression making (*Guiraldo et al., 2012*; *Hamedi Rad, Ghaffari & Safavi, 2010*), or three studies lay elastomeric impressions of control group for over 30 min after impression making (*Carvalhal et al., 2011*; *Lepe & Johnson, 1997*; *Nassar & Chow, 2015*). A total of 23 studies didn't measure the dimensions of impression or master cast before disinfection (*Abdelaziz, Hassan & Hodges, 2004*; *AlZain, 2019*; *Carvalhal et al., 2011*; *Drennon, Johnson & Powell, 1989*; *Hamedi Rad, Ghaffari & Safavi, 2010*; *Herrera & Merchant, 1986*; *Hiraguchi et al., 2012*; *Khatri et al., 2020*; *Khinnavar, Kumar & Nandeeshwar, 2015*; *Kim et al., 2008*; *Lad et al., 2015*; *Langenwalter, Aquilino & Turner, 1990*; *Lepe & Johnson, 1997*; *Lepe, Johnson & Berg, 1995*; *Lepe et al., 2002*; *Nimonkar et al., 2019*; *Queiroz et al., 2013*; *Rentzia et al., 2011*; *Rueggeberg et al., 1992*; *Saleh Saber, Abolfazli & Kohsoltani, 2010*; *Shambhu & Gujjari, 2010*; *Shetty, Kamat & Shetty, 2013*; *Silva & Salvador, 2004*; *Sinobad et al., 2014*). A total of 11 studies did not report whether multiple measurements were carried out or not (*Abdelaziz, Hassan & Hodges, 2004*; *Carvalhal et al., 2011*; *Davis & Powers, 1994*; *Guiraldo et al., 2017*; *Guiraldo et al., 2012*; *Hamedi Rad, Ghaffari & Safavi, 2010*; *Khatri et al., 2020*; *Nassar & Chow, 2015*; *Nassar et al., 2017*; *Nimonkar et al., 2019*; *Shambhu & Gujjari, 2010*). In addition, four studies did not conduct statistical analysis when evaluating detail reproduction (*Guiraldo et al., 2017*; *Guiraldo et al., 2012*; *Nassar & Chow, 2015*; *Rueggeberg et al., 1992*).

## Disinfection efficacy

A total of 13 studies evaluated disinfection efficacy of sodium hypochlorite and glutaraldehyde on dental impressions were included (*Aeran et al., 2015*; *Azevedo et al., 2019*; *Badrian et al., 2012*; *Choudhury et al., 2018*; *Doddamani, Patil & Gangadhar, 2011*; *Haralur et al., 2012*; *Jeyapalan et al., 2018*; *Mendonca et al., 2013*; *Rajendran et al., 2021*; *Rentzia et al., 2011*; *Schwartz et al., 1994*; *Taylor, Wright & Maryan, 2002*; *Westerholm2nd et al., 1992*). The efficacy of disinfectants was measured by percentage of bacterial growth inhibition *via* counting bacterial colonies after micro-organisms were transferred from undisinfected or disinfected impression surfaces to culture dishes and incubated for 24–48 h. The reductions in colony forming unit (CFU) could reflect the disinfecting

**Table 2  Results of included studies evaluating disinfection efficacy of sodium hypochlorite and glutaraldehyde on alginate impressions.**

| Study | Disinfection procedures | Percentage of bacterial growth prevention[*] | |
|---|---|---|---|
| | | ≥99.9% | <99.9% |
| *Babiker, Khalifa & Alhajj (2018)* | Immersed in 2% glutaraldehyde for 10 min | a | |
| *Badrian et al. (2012)* | Sprayed by 0.525% sodium hypochlorite for 5 min | | b(97.12%), c(99.63%), d(90.62%) |
| | Sprayed by 0.525% sodium hypochlorite for 10 min | | b(98.84%), c(99.54%), d(96.09%) |
| *Choudhury et al. (2018)* | Sprayed by 0.525% sodium hypochlorite for 5 min | | b(97.20%), c(98.10%), d(98.20%) |
| | Sprayed by 0.525% sodium hypochlorite for 10 min | | b(98.50%), c(99.00%), d(99.10%) |
| *Doddamani, Patil & Gangadhar (2011)* | Sprayed by 0.525% sodium hypochlorite for 10 min | b, e, f | |
| | Sprayed by 2% glutaraldehyde for 10 min | b, f | e(97.74%) |
| *Haralur et al. (2012)* | Sprayed by 0.525% sodium hypochlorite for 10 min | a | |
| *Rajendran et al. (2021)* | Sprayed by 2% glutaraldehyde for 10 min | a | |
| *Rentzia et al. (2011)* | Immersed in 0.525% sodium hypochlorite for 2 min | c | |
| *Schwartz et al. (1994)* | Immersed in 0.525% sodium hypochlorite for 10 min | b, c, g, h | e(98.26%) |
| *Taylor, Wright & Maryan (2002)* | Immersed in 1% sodium hypochlorite for 10 min | b | |
| *Westerholm2nd et al. (1992)* | Sprayed by 0.525% sodium hypochlorite for 10 min | a, b, e, i | |

Notes.

[*]Bacteria are represented by different letters as follows: a, human oral flora; b, *Staphylococcus aureus*; c, *Pseudomonas aeruginosa*; d, *Candida albicans*; e, *Bacillus subtilis*; f, *Streptococcus viridans*; g, *Mycobacterium bovis*; h, *Salmonella choleraesuis*; i, *Mycobacterium phlei*.

**Table 3  Results of included studies evaluating disinfection efficacy of sodium hypochlorite and glutaraldehyde to polyether and addition silicone impressions.**

| Study | Impression materials[a] | Disinfection procedures | Percentage of bacterial growth prevention[b] | |
|---|---|---|---|---|
| | | | ≥99.9% | <99.9% |
| *Aeran et al. (2015)* | P, AS | Immersed in 2% glutaraldehyde for 10 min | a | |
| *Azevedo et al. (2019)* | AS | Immersed in 1/5.25% sodium hypochlorite for 10 min | a | |
| *Jeyapalan et al. (2018)* | AS | Immersed in 1% sodium hypochlorite for 10 min | | a(99.82%) |
| | | Immersed in 2.4% glutaraldehyde for 10 min | | a(99.60%) |
| *Mendonca et al. (2013)* | AS | Sprayed by 2% glutaraldehyde for 10 min | b | c(89.13%) |

Notes.

[a]Impression materials are encoded as follows: P, polyether; AS, addition silicone.

[b]Bacteria are represented by different letters as follows : a, human oral flora; b, *Staphylococcus aureus*; c, *Bacillus atrophaeus*

efficacy of killing or inhibiting bacteria (*Aeran et al., 2015*; *Badrian et al., 2012*; *Choudhury et al., 2018*; *Doddamani, Patil & Gangadhar, 2011*; *Jeyapalan et al., 2018*; *Mendonca et al., 2013*; *Rajendran et al., 2021*; *Schwartz et al., 1994*; *Westerholm2nd et al., 1992*). Of them, six studies investigated the disinfecting efficacy of the two disinfectants on mixed oral flora (*Aeran et al., 2015*; *Azevedo et al., 2019*; *Haralur et al., 2012*; *Jeyapalan et al., 2018*; *Rajendran et al., 2021*; *Westerholm2nd et al., 1992*), while eight studies investigated the specific pathogenic micro-organisms (*Badrian et al., 2012*; *Choudhury et al., 2018*; *Doddamani, Patil & Gangadhar, 2011*; *Mendonca et al., 2013*; *Rentzia et al., 2011*; *Schwartz et al., 1994*; *Taylor, Wright & Maryan, 2002*; *Westerholm2nd et al., 1992*). The disinfection duration of all the studies was 10 min (*Aeran et al., 2015*; *Azevedo et al., 2019*; *Badrian*

**Table 4   Results of included studies evaluating surface properties of disinfected alginate impression materials.**

| Study | Disinfection procedures | | | | Surface properties evaluated[b] | | |
|---|---|---|---|---|---|---|---|
| | Type and concentration of disinfectants | Disinfection method | Disinfection duration | Control group[a] | Dimensional stability | Detail reproduction | Surface roughness |
| Babiker, Khalifa & Alhajj (2018) | 1%/5.25% sodium hypochlorite | Immersion | 5 min | NR | S | NR | NR |
| | | Spray | | | NS | NR | NR |
| Guiraldo et al. (2012) | 2% sodium hypochlorite | Spray | 15 min | NR | NS | NS | NR |
| Hamedi Rad, Ghaffari & Safavi (2010) | 5.25% sodium hypochlorite | Immersion, Spray | 8 min | WE | NS | NR | NR |
| | 2% glutaraldehyde | | | | S | NR | NR |
| Herrera & Merchant (1986) | 0.5%/1% sodium hypochlorite | Immersion | 30 min | W, A | NS | NR | NR |
| | 0.13% glutaraldehyde | | | | S | NR | NR |
| | 2% glutaraldehyde | | | | NS | NR | NR |
| Hiraguchi et al. (2012) | 0.5% sodium hypochlorite | Immersion | 15 min | NR | NS | NR | NR |
| Rentzia et al. (2011) | 0.5% sodium hypochlorite | Immersion | 30/60/90/120/ 180/240/300 s | NR | NS | NR | NR |
| Rueggeberg et al. (1992) | 0.525% sodium hypochlorite | Immersion | 10 min | NR | S | S | NR |
| | | Spray | | | NS | S | NR |
| Shambhu & Gujjari (2010) | 0.525% sodium hypochlorite | Immersion | 1/5/10 min | NR | NR | NS | NR |

**Notes.**

[a]Dental impressions of control groups were kept in air (A), water (W) or wet environment (WE). NR: Dental impressions of control groups were not reported to be kept in air, water or wet environment.

[b]Effects of disinfection on the surface properties of dental impressions are indicated as follows: S, significant difference was found when compared to control group; NS, no significant difference; NR, not reported.

Qiu et al. (2023), *PeerJ*, DOI 10.7717/peerj.14868

**Table 5** Results of included studies evaluating surface properties of disinfected polyether impression materials.

| Study | Disinfection procedures | | | | Surface properties evaluated[b] | | |
|---|---|---|---|---|---|---|---|
| | Type and concentration of disinfectants | Disinfection method | Disinfection duration | Control group[a] | Dimensional stability | Detail reproduction | Surface roughness |
| *Abdelaziz, Hassan & Hodges (2004)* | 2% glutaraldehyde | Immersion | 8 h | NR | S | NR | NS |
| *Adabo et al. (1999)* | 5.25% sodium hypochlorite | Immersion | 10 min | A | NS | NR | NR |
| | 2% glutaraldehyde | | 30 min | | NS | NR | NR |
| *AlZain (2019)* | 0.5% glutaraldehyde | Spray | 10 min | NR | NR | NR | NS |
| *Carvalhal et al. (2011)* | 0.5% sodium hypochlorite | Immersion | 5/10/20 min | W | NS | NR | NR |
| | 2% glutaraldehyde | | | | NS | NR | NR |
| | 0.5% sodium hypochlorite | | 30/60 min | | S | NR | NR |
| | 2% glutaraldehyde | | | | S | NR | NR |
| *Davis & Powers (1994)* | 2% glutaraldehyde | Immersion | 30 min/60 min | A | NS | NR | NS |
| | | | 24 h | | S | NR | NS |
| *Drennon, Johnson & Powell (1989)* | 0.25% glutaraldehyde | Spray | 10 min | NR | S | NR | NR |
| *Gounder & Vikas (2016)* | 1% sodium hypochlorite | Immersion | 30 min | W | NS | NR | NR |
| | | Immersion | 60 min | | S | NR | NR |
| | | Spray | 30/60 min | | NS | NR | NR |
| | 2% glutaraldehyde | Immersion, Spray | 30/60 min | | NS | NR | NR |
| *Guiraldo et al. (2017)* | 2% sodium hypochlorite | Immersion | 15 min | NR | S | NS | NR |
| *Herrera & Merchant (1986)* | 0.5%/1% sodium hypochlorite | Immersion | 30 min | W, A | NS | NR | NR |
| | 0.13%/2% glutaraldehyde | | | | NS | NR | NR |
| *Khatri et al. (2020)* | 3% sodium hypochlorite | Immersion | 10 min | NR | NS | NS | NR |
| | 2.45% glutaraldehyde | | | | NS | NS | NR |
| *Khinnavar, Kumar & Nandeeshwar (2015)* | 0.25% sodium hypochlorite | Immersion | 16 h | NR | S | NR | NR |
| | 2% glutaraldehyde | | | | S | NR | NR |
| *Lad et al. (2015)* | 4% sodium hypochlorite | Immersion | 10 min, | NR | NR | NR | NS |
| | 2% glutaraldehyde | | 10 | | NR | NR | NS |
| *Langenwalter, Aquilino & Turner (1990)* | 0.05% sodium hypochlorite | Immersion | 10 min | W, A | NS | NR | NR |
| | 2% glutaraldehyde | | | | NS | NR | NR |
| *Lepe, Johnson & Berg (1995)* | 2% glutaraldehyde | Immersion | 1/18 h | NR | NR | NR | NS |
| *Lepe & Johnson (1997)* | 2% glutaraldehyde | Immersion | 18 h | A | S | NR | NR |
| *Martins et al. (2017)* | 5.25% sodium hypochlorite | Immersion | 10 min | NR | NS | NR | NR |

Peer J

**Table 5** (*continued*)

| Study | Disinfection procedures | | | | Surface properties evaluated[b] | | |
|---|---|---|---|---|---|---|---|
| | Type and concentration of disinfectants | Disinfection method | Disinfection duration | Control group[a] | Dimensional stability | Detail reproduction | Surface roughness |
| *Queiroz et al. (2013)* | 2% glutaraldehyde | Immersion | 10 min | W | NS | NR | NR |
| | 0.5% sodium hypochlorite | | 10/30 min | | NR | NR | S |
| *Shetty, Kamat & Shetty (2013)* | 2% glutaraldehyde | Immersion | 10 min | NR | NR | NR | NS |
| | | | 30 min | | NR | NR | S |
| *Thouati et al. (1996)* | 5.25% sodium hypochlorite | Immersion | 30 min | NR | S | NR | NR |
| *Walker et al. (2007)* | 0.5% sodium hypochlorite | Immersion | 10 min/1 h | NR | S | NR | NR |
| *Yilmaz et al. (2007)* | 0.525% sodium hypochlorite | Immersion | 10 min | A | NS | NR | NR |
| | 2% glutaraldehyde | | | | NS | NR | NR |

**Notes.**

[a] Dental impressions of control groups were kept in air (A), water (W) or wet environment (WE). NR: Dental impressions of control groups were not reported to be kept in air, water or wet environment.

[b] Effects of disinfection on the surface properties of dental impressions are indicated as follows: S, significant difference was found when compared to control group; NS, no significant difference; NR, not reported.

**Table 6  Results of included studies evaluating surface properties of disinfected addition silicone and vinyl polyether silicone impression materials.**

| Study | Impression materials[a] | Disinfection procedures | | | | Surface properties evaluated[c] | | |
|---|---|---|---|---|---|---|---|---|
| | | Type and concentration of disinfectants | Disinfection method | Disinfection duration | Control group[b] | Dimensional stability | Detail reproduction | Surface roughness |
| Abdelaziz, Hassan & Hodges (2004) | AS | 2% glutaraldehyde | Immersion | 8 h | NR | S | NR | S |
| Adabo et al. (1999) | AS | 5.25% sodium hypochlorite | Immersion | 10 min | A | NS | NR | NR |
| | | 2% glutaraldehyde | | 30 min | | NS | NR | NR |
| AlZain (2019) | AS | 0.5% glutaraldehyde | Spray | 10 min | NR | NR | NR | NS |
| Azevedo et al. (2019) | AS | 1%/5.25% sodium hypochlorite | Immersion | 10 min | NR | NS | NR | NR |
| Carvalhal et al. (2011) | AS | 0.5% sodium hypochlorite | Immersion | 5/10/20 min | W | NS | NR | NR |
| | | 2% glutaraldehyde | | | | NS | NR | NR |
| | | 0.5% sodium hypochlorite | | 30/60 min | | S | NR | NR |
| | | 2% glutaraldehyde | | | | S | NR | NR |
| Davis & Powers (1994) | AS | 2% glutaraldehyde | Immersion | 30 min/60 min | A | NS | NR | NS |
| | | | | 24 h | | NS | NR | S |
| Drennon, Johnson & Powell (1989) | AS | 0.25% glutaraldehyde | Spray | 10 min | NR | S | NR | NR |
| Gounder & Vikas (2016) | AS | 1% sodium hypochlorite | Immersion, Spray | 30/60 min | W | NS | NR | NR |
| | | 2% glutaraldehyde | | | | NS | NR | NR |
| Guiraldo et al. (2017) | AS | 2% sodium hypochlorite | Immersion | 15 min | NR | S | NS | NR |
| Herrera & Merchant (1986) | AS | 0.5%/1% sodium hypochlorite | Immersion | 30 min | W, A | NS | NR | NR |
| | | 0.13%/2% glutaraldehyde | | | | NS | NR | NR |
| Hiraguchi et al. (2013) | AS | 2% glutaraldehyde | Immersion | 30 min/24 h | A | NS | NR | NR |
| Khatri et al. (2020) | AS, VPES | 3% sodium hypochlorite | Immersion | 10 min | NR | NS | NS | NR |
| | | 2.45% glutaraldehyde | | | | NS | NS | NR |
| Khinnavar, Kumar & Nandeeshwar (2015) | AS | 0.25% sodium hypochlorite | Immersion | 16 h | NR | S | NR | NR |
| | | 2% glutaraldehyde | | | | S | NR | NR |
| Kim et al. (2008) | AS | 6% sodium hypochlorite | Immersion | 30 min | NR | NR | NR | S |
| Lad et al. (2015) | AS | 4% sodium hypochlorite | Immersion | 10 min, 10 min | NR | NR | NR | NS |
| | | 2% glutaraldehyde | | | | NR | NR | NS |
| Langenwalter, Aquilino & Turner (1990) | AS | 0.05% sodium hypochlorite | Immersion | 10 min | W, A | NS | NR | NR |
| | | 2% glutaraldehyde | | | | NS | NR | NR |
| Lepe, Johnson & Berg (1995) | AS | 2% glutaraldehyde | Immersion | 1/18 h | NR | NR | NR | NS |
| Lepe & Johnson (1997) | AS | 2% glutaraldehyde | Immersion | 18 h | A | S | NR | NR |
| Lepe et al. (2002) | AS | 2% glutaraldehyde | Immersion | 30 min | NR | NR | NR | S |
| Martins et al. (2017) | AS | 5.25% sodium hypochlorite | Immersion | 10 min | NR | NS | NR | NR |

**Table 6** (*continued*)

| Study | Impression materials[a] | Disinfection procedures | | | | Surface properties evaluated[c] | | |
|---|---|---|---|---|---|---|---|---|
| | | Type and concentration of disinfectants | Disinfection method | Disinfection duration | Control group[b] | Dimensional stability | Detail reproduction | Surface roughness |
| *Nassar & Chow (2015)* | VPES | 2.5% glutaraldehyde | Immersion | 30 min | W | NS | NS | NR |
| *Nassar et al. (2017)* | VPES | 2.5% glutaraldehyde | Immersion | 30 min | NR | NS | NR | NR |
| *Nimonkar et al. (2019)* | AS | 1% sodium hypochlorite | Immersion | 20 min | NR | S | NR | NR |
| | | 2% glutaraldehyde | | | | S | NR | NR |
| *Piva et al. (2019)* | AS | 2% glutaraldehyde | Immersion, Spray | 10 min | NR | NS | NR | NR |
| *Queiroz et al. (2013)* | AS | 2% glutaraldehyde | Immersion | 10 min | W | NS | NR | NR |
| *Sinobad et al. (2014)* | AS | 5.25% sodium hypochlorite | Immersion | 10 min | NR | S | NR | NR |
| *Thouati et al. (1996)* | AS | 5.25% sodium hypochlorite | Immersion | 30 min | NR | S | NR | NR |
| *Walker et al. (2007)* | AS | 0.5% sodium hypochlorite | Immersion | 10 min/1 h | NR | NS | NR | NR |

**Notes.**

[a]Impression materials are encoded as follows: AS, addition silicone; VPES, vinyl polyether silicone.

[b]Dental impressions of control groups were kept in air (A), water (W) or wet environment (WE). NR: Dental impressions of control groups were not reported to be kept in air, water or wet environment.

[c]Effects of disinfection on the surface properties of dental impressions are indicated as follows: S, significant difference was found when compared to control group; NS, no significant difference; NR, not reported.

Qiu et al. (2023), *PeerJ*, DOI 10.7717/peerj.14868

**Table 7** Results of included studies evaluating surface properties of disinfected condensation silicone impression materials.

| Study | Disinfection procedures | | | | Surface properties evaluated[b] | | |
|---|---|---|---|---|---|---|---|
| | Type and concentration of disinfectants | Disinfection method | Disinfection duration | Control group[a] | Dimensional stability | Detail reproduction | Surface roughness |
| *Adabo et al. (1999)* | 5.25% sodium hypochlorite | Immersion | 10 min | A | NS | NR | NR |
| | 2% glutaraldehyde | | 30 min | | NS | NR | NR |
| *Carvalhal et al. (2011)* | 0.5% sodium hypochlorite | Immersion | 5/10/20 min | W | NS | NR | NR |
| | 2% glutaraldehyde | | | | NS | NR | NR |
| | 0.5% sodium hypochlorite | | 30/60 min | | S | NR | NR |
| | 2% glutaraldehyde | | | | S | NR | NR |
| *Guiraldo et al. (2017)* | 2% sodium hypochlorite | Immersion | 15 min | NR | S | NS | NR |
| *Lad et al. (2015)* | 4% sodium hypochlorite | Immersion | 10 min, 10 | NR | NR | NR | NS |
| | 2% glutaraldehyde | | | | NR | NR | NS |
| *Saleh Saber, Abolfazli & Kohsoltani (2010)* | 0.525% sodium hypochlorite | Spray | 10 min | NR | NS | NR | NR |
| *Silva & Salvador (2004)* | 1% sodium hypochlorite | Immersion | 10/20 min | A | NS | NR | NR |
| | 2% glutaraldehyde | | | | NS | NR | NR |
| *Sinobad et al. (2014)* | 5.25% sodium hypochlorite | Immersion | 10 min | NR | S | NR | NR |
| *Thouati et al. (1996)* | 5.25% sodium hypochlorite | Immersion | 30 min | NR | S | NR | NR |

**Notes.**

[a]Dental impressions of control groups were kept in air (A), water (W) or wet environment (WE). NR: Dental impressions of control groups were not reported to be kept in air, water or wet environment.
[b]Effects of disinfection on the surface properties of dental impressions are indicated as follows: S, significant difference was found when compared to control group; NS, no significant difference; NR, not reported.

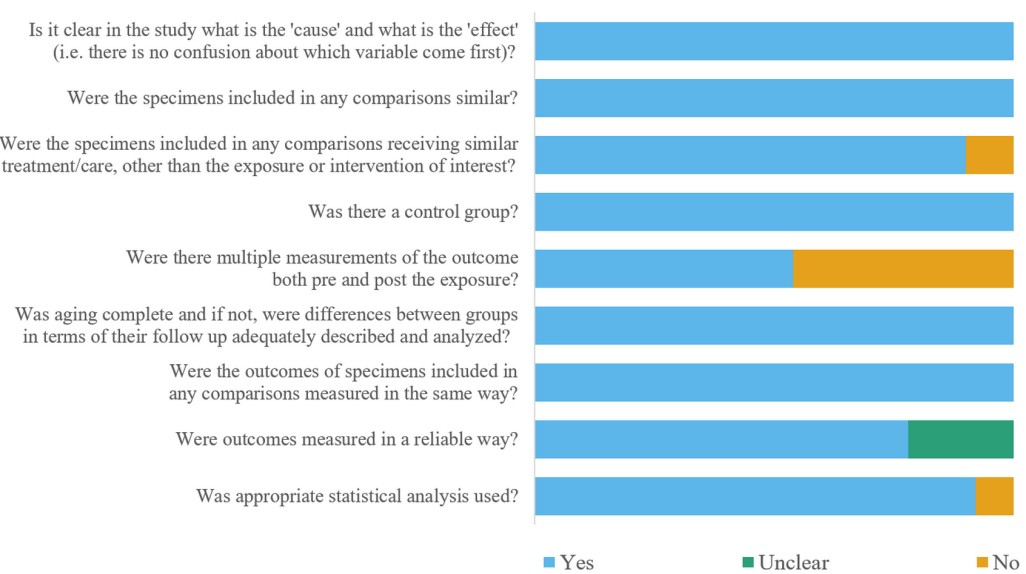

**Figure 2   Figure of the risk of bias assessment.**

*et al., 2012*; *Choudhury et al., 2018*; *Doddamani, Patil & Gangadhar, 2011*; *Haralur et al., 2012*; *Jeyapalan et al., 2018*; *Mendonca et al., 2013*; *Rajendran et al., 2021*; *Rentzia et al., 2011*; *Schwartz et al., 1994*; *Taylor, Wright & Maryan, 2002*; *Westerholm2nd et al., 1992*).

The results indicated that 10-minute disinfection with 0.5–1% sodium hypochlorite or 2% glutaraldehyde was effective against human oral micro-organisms (*Aeran et al., 2015*; *Azevedo et al., 2019*; *Badrian et al., 2012*; *Choudhury et al., 2018*; *Doddamani, Patil & Gangadhar, 2011*; *Haralur et al., 2012*; *Jeyapalan et al., 2018*; *Mendonca et al., 2013*; *Rajendran et al., 2021*; *Rentzia et al., 2011*; *Schwartz et al., 1994*; *Taylor, Wright & Maryan, 2002*; *Westerholm2nd et al., 1992*). The majority of the studies showed that 0.5–1% sodium hypochlorite or 2% glutaraldehyde achieved a vast high level reduction ($\geq$99.9%) of mixed oral flora or common oral pathogenic bacteria, including *S. aureus* and *P. aeruginosa* (*Aeran et al., 2015*; *Azevedo et al., 2019*; *Doddamani, Patil & Gangadhar, 2011*; *Haralur et al., 2012*; *Rajendran et al., 2021*; *Rentzia et al., 2011*; *Schwartz et al., 1994*; *Taylor, Wright & Maryan, 2002*; *Westerholm2nd et al., 1992*). Other studies reported this disinfection procedure could reach a higher level bacterial growth inhibition, ranging from 96.09% to 99.82% (*Badrian et al., 2012*; *Choudhury et al., 2018*; *Doddamani, Patil & Gangadhar, 2011*; *Jeyapalan et al., 2018*; *Schwartz et al., 1994*). Merely one study reported immersion disinfection with 2% glutaraldehyde was less disinfecting efficacy against *B. atrophaeus*, with a reduction percentage of 89.13% (*Mendonca et al., 2013*).

## Dimensional stability

A total of 32 studies with regard to dimensional changes were included (*Abdelaziz, Hassan & Hodges, 2004*; *Adabo et al., 1999*; *Azevedo et al., 2019*; *Babiker, Khalifa & Alhajj, 2018*; *Carvalhal et al., 2011*; *Davis & Powers, 1994*; *Drennon, Johnson & Powell, 1989*; *Gounder & Vikas, 2016*; *Guiraldo et al., 2017*; *Guiraldo et al., 2012*; *Hamedi Rad, Ghaffari & Safavi,*

*2010*; *Herrera & Merchant, 1986*; *Hiraguchi et al., 2013*; *Hiraguchi et al., 2012*; *Khatri et al., 2020*; *Khinnavar, Kumar & Nandeeshwar, 2015*; *Langenwalter, Aquilino & Turner, 1990*; *Lepe & Johnson, 1997*; *Martins et al., 2017*; *Nassar & Chow, 2015*; *Nassar et al., 2017*; *Nimonkar et al., 2019*; *Piva et al., 2019*; *Queiroz et al., 2013*). The dimensional stability of the dental impressions was evaluated by comparing measurements of the disinfected impressions or their corresponding casts with those of the undisinfected impressions or their corresponding casts. The majority of measuring devices in the studies were used with an optical microscope (*Carvalhal et al., 2011*; *Davis & Powers, 1994*; *Drennon, Johnson & Powell, 1989*; *Gounder & Vikas, 2016*; *Guiraldo et al., 2017*; *Guiraldo et al., 2012*; *Khatri et al., 2020*; *Khinnavar, Kumar & Nandeeshwar, 2015*; *Langenwalter, Aquilino & Turner, 1990*; *Lepe & Johnson, 1997*; *Martins et al., 2017*; *Nassar & Chow, 2015*; *Nassar et al., 2017*; *Nimonkar et al., 2019*; *Rentzia et al., 2011*; *Saleh Saber, Abolfazli & Kohsoltani, 2010*; *Silva & Salvador, 2004*; *Thouati et al., 1996*; *Walker et al., 2007*; *Yilmaz et al., 2007*), and the other studies adopted digital caliper (*Abdelaziz, Hassan & Hodges, 2004*; *Babiker, Khalifa & Alhajj, 2018*; *Hamedi Rad, Ghaffari & Safavi, 2010*; *Herrera & Merchant, 1986*), profile projector and digital measuring system (*Adabo et al., 1999*; *Piva et al., 2019*), camera and image analysis software (*Azevedo et al., 2019*; *Sinobad et al., 2014*), micrometer (*Hiraguchi et al., 2013*; *Rueggeberg et al., 1992*), optical comparator (*Queiroz et al., 2013*), and three-dimensional coordinate measuring system (*Hiraguchi et al., 2012*).

A majority of the included studies evaluated the dimensional changes of addition silicone impressions after disinfection (*Abdelaziz, Hassan & Hodges, 2004*; *Adabo et al., 1999*; *Azevedo et al., 2019*; *Carvalhal et al., 2011*; *Davis & Powers, 1994*; *Drennon, Johnson & Powell, 1989*; *Gounder & Vikas, 2016*; *Guiraldo et al., 2017*; *Herrera & Merchant, 1986*; *Hiraguchi et al., 2013*; *Khatri et al., 2020*; *Khinnavar, Kumar & Nandeeshwar, 2015*; *Langenwalter, Aquilino & Turner, 1990*; *Lepe & Johnson, 1997*; *Martins et al., 2017*; *Nassar & Chow, 2015*; *Nassar et al., 2017*; *Nimonkar et al., 2019*; *Piva et al., 2019*; *Queiroz et al., 2013*; *Sinobad et al., 2014*; *Thouati et al., 1996*; *Walker et al., 2007*). Among these studies, 17 studies suggested that addition silicone impression materials demonstrated the superior stability when they were disinfected by chemical disinfectants, regardless of disinfection duration and methods (*Adabo et al., 1999*; *Azevedo et al., 2019*; *Davis & Powers, 1994*; *Drennon, Johnson & Powell, 1989*; *Gounder & Vikas, 2016*; *Herrera & Merchant, 1986*; *Hiraguchi et al., 2013*; *Khatri et al., 2020*; *Khinnavar, Kumar & Nandeeshwar, 2015*; *Langenwalter, Aquilino & Turner, 1990*; *Martins et al., 2017*; *Nassar & Chow, 2015*; *Nassar et al., 2017*; *Piva et al., 2019*; *Queiroz et al., 2013*; *Thouati et al., 1996*; *Walker et al., 2007*). Contrarily, six studies reported that the dimensions of addition silicone impressions were significantly changed after disinfection (*Abdelaziz, Hassan & Hodges, 2004*; *Carvalhal et al., 2011*; *Guiraldo et al., 2017*; *Lepe & Johnson, 1997*; *Nimonkar et al., 2019*; *Sinobad et al., 2014*). However, it is noteworthy that the disinfection duration of the half studies was prolonged to more than 30 min (*Abdelaziz, Hassan & Hodges, 2004*; *Carvalhal et al., 2011*; *Lepe & Johnson, 1997*), in which one study reported that a short disinfection duration (5–20 min) did not result in significant dimensional changes of impressions (*Carvalhal et al., 2011*).

Of the eight studies evaluating condensation silicone impressions, more than half of the studies suggested that disinfection procedure could significantly affect the dimensional stability (*Adabo et al., 1999*; *Amin et al., 2009*; *Carvalhal et al., 2011*; *Guiraldo et al., 2017*; *Sabino-Silva, Jardim & Siqueira, 2020*; *Saleh Saber, Abolfazli & Kohsoltani, 2010*; *Sinobad et al., 2014*; *Thouati et al., 1996*). Five studies reported that the use of sodium hypochlorite caused significant dimensional changes of condensation silicone impressions (*Carvalhal et al., 2011*; *Guiraldo et al., 2017*; *Saleh Saber, Abolfazli & Kohsoltani, 2010*; *Sinobad et al., 2014*; *Thouati et al., 1996*). In addition, one study reported 2% glutaraldehyde immersion similarly caused adverse effects on the impressions (*Carvalhal et al., 2011*). Other 2 studies reported that the condensation silicone impressions were not significantly affected after disinfection (*Adabo et al., 1999*; *Silva & Salvador, 2004*).

Of the 19 studies investigating polyether impressions, 13 studies investigated sodium hypochlorite (*Adabo et al., 1999*; *Carvalhal et al., 2011*; *Gounder & Vikas, 2016*; *Guiraldo et al., 2017*; *Herrera & Merchant, 1986*; *Khatri et al., 2020*; *Khinnavar, Kumar & Nandeeshwar, 2015*; *Langenwalter, Aquilino & Turner, 1990*; *Martins et al., 2017*; *Queiroz et al., 2013*; *Thouati et al., 1996*; *Walker et al., 2007*; *Yilmaz et al., 2007*), and 14 studies investigated glutaraldehyde (*Abdelaziz, Hassan & Hodges, 2004*; *Adabo et al., 1999*; *Carvalhal et al., 2011*; *Davis & Powers, 1994*; *Drennon, Johnson & Powell, 1989*; *Gounder & Vikas, 2016*; *Herrera & Merchant, 1986*; *Khatri et al., 2020*; *Khinnavar, Kumar & Nandeeshwar, 2015*; *Langenwalter, Aquilino & Turner, 1990*; *Lepe & Johnson, 1997*; *Nassar & Chow, 2015*; *Nassar et al., 2017*; *Yilmaz et al., 2007*). With regard to sodium hypochlorite, seven studies indicated sodium hypochlorite didn't provoke significant dimensional changes of the impressions or casts (*Adabo et al., 1999*; *Herrera & Merchant, 1986*; *Khatri et al., 2020*; *Langenwalter, Aquilino & Turner, 1990*; *Martins et al., 2017*; *Queiroz et al., 2013*; *Yilmaz et al., 2007*). However, six other studies reported the adverse effect of sodium hypochlorite on the dimensional stability of polyether impressions, at the concentration ranging from 0.5% to 5.25% (*Carvalhal et al., 2011*; *Gounder & Vikas, 2016*; *Guiraldo et al., 2017*; *Khinnavar, Kumar & Nandeeshwar, 2015*; *Thouati et al., 1996*; *Walker et al., 2007*). As for glutaraldehyde, most studies demonstrated that it could not cause significant changes in dimensions (*Adabo et al., 1999*; *Davis & Powers, 1994*; *Drennon, Johnson & Powell, 1989*; *Gounder & Vikas, 2016*; *Herrera & Merchant, 1986*; *Khatri et al., 2020*; *Langenwalter, Aquilino & Turner, 1990*; *Nassar & Chow, 2015*; *Nassar et al., 2017*; *Yilmaz et al., 2007*), while other four studies revealed the adverse effect of 2% glutaraldehyde on the impressions with a disinfection period among 30 min to 18 h (*Abdelaziz, Hassan & Hodges, 2004*; *Carvalhal et al., 2011*; *Khinnavar, Kumar & Nandeeshwar, 2015*; *Lepe & Johnson, 1997*). In addition, seven studies reported that dimensions of polyether impressions were significantly affected after immersion disinfection for over 30 min (*Abdelaziz, Hassan & Hodges, 2004*; *Carvalhal et al., 2011*; *Davis & Powers, 1994*; *Gounder & Vikas, 2016*; *Khinnavar, Kumar & Nandeeshwar, 2015*; *Lepe & Johnson, 1997*; *Walker et al., 2007*). Furthermore, two studies pointed out that dimensional changes of polyether impressions were increased in pace with the prolongation of disinfection duration (*Davis & Powers, 1994*; *Gounder & Vikas, 2016*).

Of the seven studies investigating alginate impressions, five of them pointed out the innocuous effect of sodium hypochlorite on dimensional stability of alginate impressions,

with disinfection duration ranging from 5 to 30 min (*Guiraldo et al., 2012*; *Hamedi Rad, Ghaffari & Safavi, 2010*; *Herrera & Merchant, 1986*; *Hiraguchi et al., 2012*; *Rentzia et al., 2011*). While the other two studies reported the significant difference was detected between immersion and spray disinfection methods in two experimental groups separately (*Babiker, Khalifa & Alhajj, 2018*; *Rueggeberg et al., 1992*). In addition, one study reported that 2% glutaraldehyde immersion caused significant dimensional changes while 0.13% glutaraldehyde did not (*Herrera & Merchant, 1986*).

## Detail reproduction

Six included studies investigated the ability of the dental impressions to reproduce tiny details after disinfection (*Guiraldo et al., 2017*; *Guiraldo et al., 2012*; *Khatri et al., 2020*; *Nassar & Chow, 2015*; *Rueggeberg et al., 1992*; *Shambhu & Gujjari, 2010*). Three studies examined whether the tiny details were completely reproduced on the dental impressions or casts when evaluating detail reproduction (*Guiraldo et al., 2017*; *Guiraldo et al., 2012*; *Rueggeberg et al., 1992*), while the other three studies assessed and ranked the reproduced details (*Khatri et al., 2020*; *Nassar & Chow, 2015*; *Shambhu & Gujjari, 2010*). One study reported that the detail reproduction (a width of 50-$\mu$m line) of alginate impressions was diminished, and a width of 75-$\mu$m line could be detected after 10-minute disinfection of 0.525% sodium hypochlorite using immersion or spray disinfection method (*Rueggeberg et al., 1992*). Instead, the other two studies showed that no detectable significant changes in detail reproduction were caused by using sodium hypochlorite at the concentration of 0.525% or 2% (*Guiraldo et al., 2012*; *Shambhu & Gujjari, 2010*). As for elastomeric impression materials, three studies reported disinfection agents could not reduce surface detail reproducibility, regardless of sodium hypochlorite and glutaraldehyde used (*Guiraldo et al., 2012*; *Khatri et al., 2020*; *Nassar & Chow, 2015*).

## Wettability

The surface wettability of impressions was assessed in eight studies, by measuring the magnitude of the contact angle on the surface of impression before and after disinfection (*Abdelaziz, Hassan & Hodges, 2004*; *AlZain, 2019*; *Davis & Powers, 1994*; *Kim et al., 2008*; *Lad et al., 2015*; *Lepe, Johnson & Berg, 1995*; *Lepe et al., 2002*; *Shetty, Kamat & Shetty, 2013*). Of the seven studies investigating polyether impressions, five studies reported that glutaraldehyde could not affect the wettability of the impression surfaces (*Abdelaziz, Hassan & Hodges, 2004*; *AlZain, 2019*; *Davis & Powers, 1994*; *Lad et al., 2015*; *Lepe, Johnson & Berg, 1995*). On the contrary, one study reported the 30-minute immersion of 2% glutaraldehyde solution significantly decreased the wettability of the impression surfaces (*Lepe et al., 2002*), and another study reported that both sodium hypochlorite and glutaraldehyde were possible to affect the wettability (*Shetty, Kamat & Shetty, 2013*). As for addition silicone, four of seven studies demonstrated glutaraldehyde could significantly reduce the wettability (*Abdelaziz, Hassan & Hodges, 2004*; *Davis & Powers, 1994*; *Kim et al., 2008*; *Lepe et al., 2002*), while one study reported that 0.5% glutaraldehyde improved the wettability of addition silicone impressions effectively (*AlZain, 2019*). Besides, three studies reported that the wettability of impression surfaces could not be affected by glutaraldehyde and sodium

hypochlorite (*AlZain, 2019*; *Lad et al., 2015*; *Lepe, Johnson & Berg, 1995*). In addition, one study reported the wettability of condensation silicone impressions was not affected by 10-minute immersion disinfection of 4% sodium hypochlorite or 2% glutaraldehyde (*Lad et al., 2015*).

## DISCUSSIONS

Alginate, polyether, addition silicone and condensation silicone are four types of dental impression materials commonly used in dental clinics. Alginates are easy to use, well tolerated by patients, and excellent for primary prosthetic and orthodontic (*Cervino et al., 2018*). They come in the form of a powder to be mixed with water in appropriate doses (*Cervino et al., 2018*; *Donovan & Chee, 2004*). Once mixed, the alginate turns into a soft paste, and finally forms a gel within 2–5 min through a chemical irreversible reaction (*Cervino et al., 2018*; *Donovan & Chee, 2004*). The accuracy of the alginate impressions deteriorates over time because of water evaporation, thus immediate pouring of alginate impressions provides the highest accuracy regarding teeth and tissues (*Garrofé et al., 2015*; *Guiraldo et al., 2015*). Polyether, addition silicone and condensation silicone are elastomer impression materials, and most of them are provided as base/catalyst systems (*Chee & Donovan, 1992*; *Punj, Bompolaki & Garaicoa, 2017*). When used, they are entirely mixed by using some type of auto-mix system or hand mixing (*Chee & Donovan, 1992*; *Donovan & Chee, 2004*; *Punj, Bompolaki & Garaicoa, 2017*). Afterwards the polymerization reaction completes within several min (*Donovan & Chee, 2004*; *Punj, Bompolaki & Garaicoa, 2017*). The reaction for polyether impression materials is *via* cationic polymerization by opening of the reactive ethylene imine terminal rings to unite molecules without by-product formation (*Sakaguchi & Powers, 2012*). For addition silicone impression materials, the polymerization involves hydrosilane-terminated molecules reacting with siloxane oligomers with vinyl end groups and a platinum catalyst (*Sakaguchi & Powers, 2012*). Besides, the polymerization of condensation silicone impression materials occurs through a cross link between the terminal connections of the silicone polymer and an alkylic silicate (*Silva & Salvador, 2004*).

To obtain biologically, functionally, esthetically acceptable dental restorations, the impression materials should record the dentition and its neighboring oral tissues and transfer to the cast accurately (*Karaman, Oztekin & Tekin, 2020*; *Perakis, Belser & Magne, 2004*; *Piva et al., 2019*). Since disinfection is necessary for impressions to minimize the risk of disease transmission, surface properties should not be affected during disinfection procedures (*Amalan, Ginjupalli & Upadhya, 2013*). This systematic review comprehensively assessed the disinfection efficacy of sodium hypochlorite and glutaraldehyde, as well as the influence of different disinfectants and disinfection procedures on the surface properties of dental impressions, including dimensional stability, detail production and wettability.

### Disinfection efficacy

Sodium hypochlorite and glutaraldehyde are widely used in dental clinics and dental laboratories, owing to their high antimicrobial efficacy (*Carvalhal et al., 2011*; *Guiraldo*

*et al., 2017*; *Guiraldo et al., 2012*; *Khinnavar, Kumar & Nandeeshwar, 2015*; *Walker et al., 2007*). Sodium hypochlorite is effective against a broad spectrum of micro-organisms including bacteria and their spores, viruses and fungi, as well as HIV and hepatitis B virus (*Guiraldo et al., 2012*; *Rentzia et al., 2011*). Glutaraldehyde is considered a high-level disinfectant, which could eliminate some spores, the bacillus responsible for tuberculosis, vegetative bacteria, fungi, and viruses (*Guiraldo et al., 2017*; *Guiraldo et al., 2012*).

There are several hundred species of bacteria existing in the oral biofilms, and *S. aureus*, *P. aeruginosa*, *B. atrophaeus* are the most common species (*Doddamani, Patil & Gangadhar, 2011*; *Schwartz et al., 1994*; *Westerholm2nd et al., 1992*). *S. aureus* has been reported to be the source of numerous oral infections, including angular cheilitis, parotitis, and staphylococcal mucositis (*Bagg et al., 1995*; *MacFarlane & Helnarska, 1976*). Also, it may cause endocarditis and postoperative infections (*Doddamani, Patil & Gangadhar, 2011*). *P. aeruginosa* are associated with many lung diseases, including healthcare-associated pneumonia, chronic obstructive pulmonary disease, and cystic fibrosis (*Riquelme et al., 2020*; *Winstanley, O'Brien & Brockhurst, 2016*). As for *B. atrophaeus*, it has been isolated from individuals suffering from septicemia, meningitis, endocarditis, and pneumonia, wound infections, and other suppurative lesions (*Doddamani, Patil & Gangadhar, 2011*). These species of bacteria are ordinarily nonpathogenic but can cause diseases on occasions, thus they are appropriate to be used *in vitro* studies because of their less hazardousness (*Doddamani, Patil & Gangadhar, 2011*; *Schwartz et al., 1994*; *Westerholm2nd et al., 1992*).

Three studies reported that the two disinfectants, sodium hypochlorite and glutaraldehyde, could significantly inhibit bacterial growth at least a 4-$\log_{10}$ (99.99%) reduction in CFU counts after disinfection (*Doddamani, Patil & Gangadhar, 2011*; *Schwartz et al., 1994*; *Westerholm2nd et al., 1992*). Included studies have proved that most of the micro-organisms and potential pathogenic bacteria could be inactivated after 10-minute disinfection with 0.5–1% sodium hypochlorite or 2% glutaraldehyde, regardless of mixed oral flora or single bacterium (*Aeran et al., 2015*; *Azevedo et al., 2019*; *Badrian et al., 2012*; *Choudhury et al., 2018*; *Haralur et al., 2012*; *Jeyapalan et al., 2018*; *Mendonca et al., 2013*; *Rajendran et al., 2021*; *Rentzia et al., 2011*; *Taylor, Wright & Maryan, 2002*). It could be considered effective to inactivate bacteria on the dental impressions and minimize the risk of cross infection.

In addition, few studies focused the disinfection of dental impressions against viruses, so no concrete conclusions can be drawn. As far as available studies concerned, sodium hypochlorite and glutaraldehyde are effective to eliminate common infectious viruses in oral cavity (*Guiraldo et al., 2017*; *Guiraldo et al., 2012*; *Rentzia et al., 2011*). The medical instruments which carried danger of harbouring HIV or hepatitis viruses could be sterilized *via* chemical methods, provided all surfaces of instruments are in full contact with the disinfectants (*Adler-Storthz et al., 1983*; *Cairns, 2000*). The HIV or hepatitis B and C viruses could be inactivated by 10-minute disinfection using 0.5% sodium hypochlorite or 2% alkaline buffered glutaraldehyde (*Adler-Storthz et al., 1983*; *Cairns, 2000*). Meanwhile, a recent study reported 0.1% sodium hypochlorite or 0.5–2% glutaraldehyde could efficiently inactivate coronaviruses (*Fadaei, 2021*).

## Dimensional stability

It is widely known that alginate impressions have the propensity to absorb water because of the different osmotic pressure between the impression and the disinfection solution (*Hamedi Rad, Ghaffari & Safavi, 2010*; *Hiraguchi et al., 2012*; *Rueggeberg et al., 1992*). Meanwhile, the amount of water absorption varies with different concentrations and different kinds of disinfectants (*Guiraldo et al., 2012*). This might be attributed to the expansion of alginate during disinfection, so the restriction of immersion period is necessary (*Hamedi Rad, Ghaffari & Safavi, 2010*). *Amin et al. (2009)* and *Rentzia et al. (2011)* reported that, the syneresis prior to pouring the stone casts and the imbibition upon pouring the impressions with dental stone could affect the accuracy to some degree. However, immersion disinfection in sodium hypochlorite in this review was less likely to cause significant changes in dimensions (*Guiraldo et al., 2012*; *Hamedi Rad, Ghaffari & Safavi, 2010*; *Herrera & Merchant, 1986*; *Hiraguchi et al., 2012*; *Rentzia et al., 2011*). In addition, *Hiraguchi et al. (2012)* reported 0.5% sodium hypochlorite immersion resulted in a cast with an excellent dimension stability when compared to that of the control group. This could be explained by the fact that the solution of sodium hypochlorite is a strong electrolyte with monovalent metallic ions and the coexistence of electrolytes could cause the contraction of alginate polymer segments in aqueous media, offsetting water adsorption expansion (*Saito, Ichimaru & Araki, 1998*). *Hamedi Rad, Ghaffari & Safavi (2010)* and *Khatri et al. (2020)* reported that the distortion of the cast in some dimensions was detected after glutaraldehyde disinfection. This might be the reason that glutaraldehyde is not an electrolyte (*Saito, Ichimaru & Araki, 1998*). In addition, *Babiker, Khalifa & Alhajj (2018)* and *Rueggeberg et al. (1992)* reported significant dimensional changes resulted from the immersion disinfection method using sodium hypochlorite, while spray disinfection method using the same disinfectant at the same concentration didn't cause significant dimensional changes. As spray disinfection could limit the exposure of impressions in the wet environment, it seems to be less influential in dimensions of alginate impressions (*Babiker, Khalifa & Alhajj, 2018*; *Hamedi Rad, Ghaffari & Safavi, 2010*; *Rueggeberg et al., 1992*). Based on the included studies, spray disinfection method should be strongly recommended for alginate impressions (*Babiker, Khalifa & Alhajj, 2018*; *Rueggeberg et al., 1992*).

Polyether materials are considered to be hydrophilic (*Adabo et al., 1999*; *Davis & Powers, 1994*; *Gounder & Vikas, 2016*; *Guiraldo et al., 2018*; *Guiraldo et al., 2017*; *Walker et al., 2007*), because they contain the carbonyl (C =O) and ether (C-O-C) groups that attract and interact chemically with water molecules through hydrogen (*Guiraldo et al., 2018*; *Guiraldo et al., 2017*). Hence, polyether impressions are not suitable for disinfection for a prolonged immersion disinfection duration, especially over 30 min (*Abdelaziz, Hassan & Hodges, 2004*; *Carvalhal et al., 2011*; *Davis & Powers, 1994*; *Gounder & Vikas, 2016*; *Khinnavar, Kumar & Nandeeshwar, 2015*; *Lepe & Johnson, 1997*; *Walker et al., 2007*). Glutaraldehyde causing less distortion of dental impression seemed to be more appropriate for the disinfection of polyether impression than sodium hypochlorite (*Carvalhal et al., 2011*; *Gounder & Vikas, 2016*; *Guiraldo et al., 2017*; *Khinnavar, Kumar & Nandeeshwar, 2015*; *Thouati et al., 1996*; *Walker et al., 2007*). This might be explained by the fact that the

dimensional changes might be caused by reaction of highly reactive chlorine compound of sodium hypochlorite with sulfonic ether of polyether, which could interfere with the polymerization reaction and produce distortion (*Gounder & Vikas, 2016*; *Thouati et al., 1996*). Therefore, glutaraldehyde is strongly recommended to be used for disinfecting the polyether impressions.

Addition silicone impressions seems to be the most accurate materials in comparison with others (*Amin et al., 2009*; *Gounder & Vikas, 2016*; *Guiraldo et al., 2018*; *Guiraldo et al., 2017*; *Khinnavar, Kumar & Nandeeshwar, 2015*). Traditional addition silicone is considered to be hydrophobic (*Davis & Powers, 1994*; *Khatri et al., 2020*; *Nimonkar et al., 2019*; *Queiroz et al., 2013*), which highly resists to aqueous disinfectants regardless of the exposure of disinfection duration (*Khatri et al., 2020*). Noticeably, *Davis & Powers (1994)* and *Hiraguchi et al. (2013)* pointed out that addition silicone impressions remained dimensionally accurate even after immersion for 24 h. However, nowadays, in consideration of improving the ability to reproduce details, some addition silicone impression materials are added with surfactants, which could increase the sorption of water when dental impressions are immersed for a longer period (*Gounder & Vikas, 2016*). Thus, prolonged disinfection duration are not clinically recommended.

Vinyl polyether silicone (VPES) has a different composition from other elastomeric impression materials as it combines vinyl polysiloxane (VPS) and polyether (PE), (*Nassar et al., 2017*) so it can take advantage of the properties of both PVS and PE (*Khatri et al., 2020*). This review suggested that disinfection could not affect dimensional stability of VPES, because it did not contain surfactants regardless of its intrinsic hydrophilicity (*Khatri et al., 2020*; *Nassar & Chow, 2015*; *Nassar et al., 2017*). However, whether longer disinfection duration could affect dimensional stability is uncertain. Further studies are needed in future (*Nassar & Chow, 2015*).

Condensation silicone impressions presented less accurate results in dimensional stability in comparison with other dental elastomeric impression materials (*Amin et al., 2009*; *Carvalhal et al., 2011*; *Guiraldo et al., 2018*; *Guiraldo et al., 2017*), which might be attributed to the inherent property of the condensation silicone rather than to the disinfection (*Amin et al., 2009*). Owing to its hydrophobic nature, condensation silicone impressions are less susceptible to water sorption and react with disinfectants (*Guiraldo et al., 2017*). Besides, the polymerization reaction of elastomeric compounds results in the formation of a three-dimensional net, and ethyl alcohol is formed as a by-product (*Silva & Salvador, 2004*). Volume reduction due to the cross link and alcohol evaporation made condensation silicone impressions exhibit some degrees of contraction, sometimes uncontrollably, during polymerization (*Amin et al., 2009*; *Guiraldo et al., 2018*; *Guiraldo et al., 2017*; *Saleh Saber, Abolfazli & Kohsoltani, 2010*; *Silva & Salvador, 2004*; *Sinobad et al., 2014*). As some studies suggested, immersion disinfection inhibited the vaporization of alcohol and improved the dimensional accuracy (*Guiraldo et al., 2017*; *Saleh Saber, Abolfazli & Kohsoltani, 2010*; *Silva & Salvador, 2004*). Even though significant changes between experimental and control groups were detected, *Carvalhal et al. (2011)*, *Saleh Saber, Abolfazli & Kohsoltani (2010)* and *Thouati et al. (1996)* considered that these changes

were not sufficient to cause critical distortions of denture restorations, and the disinfection procedures were clinically feasible.

Based on above results, spray disinfection with 0.5% sodium hypochlorite, left for 10 min, for alginate impressions (*Babiker, Khalifa & Alhajj, 2018*; *Guiraldo et al., 2012*; *Hamedi Rad, Ghaffari & Safavi, 2010*; *Rueggeberg et al., 1992*) and 10-minute immersion disinfection with 0.5% sodium hypochlorite or 2% glutaraldehyde for elastomeric impressions (*Adabo et al., 1999*; *Azevedo et al., 2019*; *Carvalhal et al., 2011*; *Davis & Powers, 1994*; *Gounder & Vikas, 2016*; *Herrera & Merchant, 1986*; *Hiraguchi et al., 2013*; *Khatri et al., 2020*; *Langenwalter, Aquilino & Turner, 1990*; *Martins et al., 2017*; *Nassar & Chow, 2015*; *Nassar et al., 2017*; *Piva et al., 2019*; *Queiroz et al., 2013*; *Yilmaz et al., 2007*) did not cause adverse effect on dimensional stability, which could be considered appropriate for disinfecting impressions. Since 10-minute disinfection was effective to inhibit pathogenic micro-organisms and minimize the risk of cross infection (*Abdelaziz, Hassan & Hodges, 2004*; *Aeran et al., 2015*; *Azevedo et al., 2019*; *Doddamani, Patil & Gangadhar, 2011*; *Haralur et al., 2012*; *Mendonca et al., 2013*; *Rajendran et al., 2021*; *Rentzia et al., 2011*; *Schwartz et al., 1994*; *Taylor, Wright & Maryan, 2002*; *Westerholm2nd et al., 1992*), prolonged disinfection duration was not necessary owing to provoking potential adverse effects (*Abdelaziz, Hassan & Hodges, 2004*; *Carvalhal et al., 2011*; *Davis & Powers, 1994*; *Gounder & Vikas, 2016*; *Khinnavar, Kumar & Nandeeshwar, 2015*; *Lepe & Johnson, 1997*; *Walker et al., 2007*).

## Detail reproduction and wettability

The detail reproduction is mainly influenced by the compatibility between dental stone and impression materials (*Amalan, Ginjupalli & Upadhya, 2013*; *Hutchings et al., 1996*). Sodium hypochlorite could increase the alkalinity of the alginate impression materials, which might alter its compatibility with dental stone and affect the detail reproduction to some degree (*Amalan, Ginjupalli & Upadhya, 2013*; *Hutchings et al., 1996*). Numerous previous publications demonstrated that disinfection could not affect the detail reproduction of elastomeric impression materials, no matter what disinfectants and disinfection method were used (*Guiraldo et al., 2017*; *Guiraldo et al., 2012*; *Khatri et al., 2020*; *Nassar & Chow, 2015*; *Shambhu & Gujjari, 2010*).

With regard to wettability, addition silicone impressions seemed to be more sensitive to disinfection (*Abdelaziz, Hassan & Hodges, 2004*; *Davis & Powers, 1994*; *Kim et al., 2008*; *Lepe et al., 2002*). As previously mentioned, the addition of surfactants into silicone impressions was to improve detail reproduction (*Gounder & Vikas, 2016*), but the dilution and destruction of the surfactant during disinfection procedures would make them less wettable (*Lepe, Johnson & Berg, 1995*). Because of the natural hydrophilicity of polyether impressions, its surface wettability would not be easily affected (*Abdelaziz, Hassan & Hodges, 2004*; *AlZain, 2019*; *Davis & Powers, 1994*; *Lad et al., 2015*; *Lepe, Johnson & Berg, 1995*). As for condensation silicone, it was the least hydrophilic and wettable in all three impression materials, which would not be affected by disinfection, either (*Lad et al., 2015*).

**Table 8** Recommended disinfection methods for different dental impression materials.

| Impression material | Disinfection method | Disinfectant | Disinfection duration |
|---|---|---|---|
| Alginate | Spray disinfection | 0.5% sodium hypochlorite | 10 min |
| Polyether | Immersion disinfection | 2% glutaraldehyde | 10 min |
| Addition silicone or condensation silicone | Immersion disinfection | 0.5% sodium hypochlorite or 2% glutaraldehyde | 10 min |

In general, common disinfection procedures would not adversely affect the detail reproduction of the dental impressions. The wettability of the addition silicone impressions might be decreased after disinfection, while other impression materials are unaffected.

## Limitations

This review merely analyzed two most commonly used disinfectants, sodium hypochlorite and glutaraldehyde. Other chemical disinfectants, such as chlorhexidine, iodophor and peracetic acid, should be further studied in future.

## CONCLUSIONS

Based on the systematic review, the following conclusions could be drawn: 0.5–1% sodium hypochlorite and 2% glutaraldehyde not only could inactivate oral flora and common oral pathogenic bacteria, but also could not alter the surface properties of dental impression materials within 30 min of disinfection duration, except for the wettability of addition silicone impressions and dimensional stability of condensation silicone impressions.

Therefore, the disinfection methods for dental impression materials which strongly recommended by this study are listed in Table 8.

### Funding

The authors received no funding for this work.

### Competing Interests

The authors declare there are no competing interests.

### Author Contributions

- Yuan Qiu conceived and designed the experiments, performed the experiments, analyzed the data, prepared figures and/or tables, and approved the final draft.
- Jiawei Xu performed the experiments, analyzed the data, prepared figures and/or tables, and approved the final draft.
- Yuedan Xu performed the experiments, analyzed the data, prepared figures and/or tables, and approved the final draft.
- Zhiwei Shi analyzed the data, prepared figures and/or tables, and approved the final draft.

- Yinlin Wang analyzed the data, prepared figures and/or tables, and approved the final draft.
- Ling Zhang conceived and designed the experiments, authored or reviewed drafts of the article, and approved the final draft.
- Baiping Fu conceived and designed the experiments, authored or reviewed drafts of the article, and approved the final draft.

## Data Availability

This study is a systematic review, and all the raw data are listed in the Tables.

## Supplemental Information

Supplemental information for this article can be found online at http://dx.doi.org/10.7717/peerj.14868#supplemental-information.

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
