# Peer review of "Disinfection efficacy of sodium hypochlorite and glutaraldehyde and their effects on the dimensional stability and surface properties of dental impressions: a systematic review"

_PeerJ, doi:10.7717/peerj.14868_

## Round 0.1 · original submission · Major Revisions

Please address the issues mentioned by our reviewers (especially #1 and #2) . Reviewer #3's comments are not very substantive: I am afraid this reviewer unwittingly failed to check our requirements regarding systematic reviews, and that a mismatch in expectations unduly colored their assessment.

Reviewer 1 ·

Basic reporting

This systematic review evaluated the disinfection efficacy of the two most frequently used disinfectants, namely sodium hypochlorite and glutaraldehyde, and their effects on the surface properties of four different dental impression materials. In general, the manuscript is very difficult to follow mainly due to the listing of multiple studies on four different impression materials in a mixed order in Tables 2 and 3. The authors do not clearly provide significant insights into the disinfection conditions for each dental material. Collectively, the contribution of this work to establish standardized disinfection procedures for dental impressions is not clear.

Specific concerns
Please clearly separate and describe the effects on the four different dental materials studied in the abstract.

Introduction is written as though intended for basic research on prosthodontics. As the PeerJ audience may be more general, it would be worth expanding the introduction to include information on the importance of oral microorganisms. In particular, Staphylococcus aureus, Pseudomonas aeruginosa, and Bacillus atrophaeus, which are discussed in this study, need to be explained.

In addition, the authors need to provide basic information on the four types of dental impression materials.

The author states in the discussion the need to re-evaluate the effects of disinfectants on the surface properties of dental impressions following the disinfection guidelines recommended by the ADA; however, this would be easier to understand if clearly stated in the introduction first.

In L54-56, the presence of microorganisms other than bacteria, including viruses, is mentioned but not considered in results and discussion.

A table summarizing the authors’ recommended disinfection conditions would enhance the significance of this review.

Experimental design

No comment.

Validity of the findings

No comment.

Additional comments

No comment.

Reviewer 2 ·

Basic reporting

Regarding the writing of the article, although I am not the right person to judge the construction of sentences in English, I think it is written in an understandable and enlightening way. In this regard, I do not find any flaws.
The review is well written and explained, with a logical and traditional sequence in the theme in which it is inserted.
Congratulations on the systematic review.

Experimental design

All experimental work seems to be adequate and respects all the rules established for this kind of publication. In fact, this theme and this work comes to give a more assertive answer in relation to this controversial theme, in which not all authors would agree. Therefore, it is also important to mention the importance of this answer for the day-to-day clinical work in this very important field of Dentistry.

Validity of the findings

Despite the fact that we are currently witnessing a paradigm shift in dental impressions, namely with the great development of intraoral scanners, we are still forced to accept the reality that impression materials cannot disappear completely. Therefore, its disinfection is absolutely essential.
Regarding the data published and written in the work, I think they are of above average quality, with a good degree of detail, with robustness, easily supporting the conclusions they reached.

Additional comments

Please put the end point after the references in each sentence. It doesn't make sense the way they're written.

Reviewer 3 ·

Basic reporting

This manuscript is a review article, but the structure is arranged for the standard sections in this journal.
Additionally, the writing often lacks clarity and sharpness, and some parts are poorly organized. Many references are cited in the text, which causes a difficulty to read. References are also shown in Tables 2 and 3, so redundant sentences should be improved.

Experimental design

This manuscript only reviews previous reports without additional experimental data, and contributes little to filling the knowledge gaps. Also, in the conclusion section, authors proposed concentrations of disinfectants and the most excellent impression material, but the reasons are unclear.

Validity of the findings

Since no experimental result is present in this manuscript, I cannot describe any comments.

Additional comments

Although this manuscript is a review article and also shows a "review" in title,the style is not suitable for the review article in this journal. Please confirm the style of this manuscript.

---

## Round 0.2 · Minor Revisions

Please address the minor issues highlighted by reviewer #1

Reviewer 1 ·

Basic reporting

The authors have satisfactorily addressed the concerns. However, the authors need some modifications listed below.

1. P8 L53: Spirochetes belong to bacteria.
2. P23 L400: Please revise references (8, 14, 32-34).
3. P23 L406-414: Please change the bacterial name to italics. Genus name can be abbreviated.
4. Table 2: Please delete g in footnote, because g is missing in Table.
5. Table 3: Please delete c-f, and h-j in footnote, because these are missing in Table.

Experimental design

no comment

Validity of the findings

no comment

Additional comments

no comment

---

## Round 0.3 · accepted · Accept

Thank you for performing the last minor changes! I am glad to accept your paper for publication.